# Predictors and trends of Caesarean section and breastfeeding in the Eastern Mediterranean region: Data from the cross-sectional Cyprus Women's Health Research (COHERE) Initiative

Bethan Swift[1,2], Bahar Taneri[3,4,5], Ilgin Cagnan[3,4], Christian M. Becker[1], Krina T. Zondervan[1,2], Maria A. Quigley[6], Nilufer Rahmioglu[1,2,4]*

1 Oxford Endometriosis CaRe Centre, Nuffield Department of Women's and Reproductive Health, University of Oxford, Oxford, United Kingdom, 2 Wellcome Centre for Human Genetics, University of Oxford, Oxford, United Kingdom, 3 Faculty of Arts and Sciences, Department of Biological Sciences, Eastern Mediterranean University, Famagusta, Northern Cyprus, 4 Cyprus Women's Health Research Society (CoHERS), Nicosia, Northern Cyprus, 5 Institute for Public Health Genomics (IPHG), Department of Genetics and Cell Biology, Research Institute GROW, Faculty of Health, Medicine & Life Sciences, University of Maastricht, Maastricht, The Netherlands, 6 National Perinatal Epidemiology Unit, Nuffield Department of Population Health, University of Oxford, Oxford, United Kingdom

* nilufer.rahmioglu@wrh.ox.ac.uk

## Abstract

### Introduction

Caesarean section (C-section) is a life-saving procedure when medically indicated but unmet need and overuse can add to avoidable morbidity and mortality. It is not clear whether C-section has a negative impact on breastfeeding and there is limited data available on rates of C-section or breastfeeding from Northern Cyprus, an emerging region in Europe. This study aimed to investigate prevalence, trends and associations of C-section and breastfeeding in this population.

### Methods

Using self-reported data from the representative Cyprus Women's Health Research (COHERE) Initiative, we used 2,836 first pregnancies to describe trends in C-section and breastfeeding between 1981 and 2017. Using modified Poisson regression, we examined the relationship between year of pregnancy and C-section and breastfeeding, as well as the association between C-section and breastfeeding prevalence and duration.

### Results

C-section prevalence in first pregnancies increased from 11.1% in 1981 to 72.5% in 2017 with a relative risk of 2.60 (95%CI; 2.14–2.15) of babies being delivered by C-section after 2005 compared to before 1995, after full adjustment for demographic and maternal medical and pregnancy related factors. Prevalence of ever breastfeeding remained steady

**Data Availability Statement:** There are ethical restrictions on sharing the data publicly as not all study participants consented to data being made

publicly available. A subset of anonymous data can be requested from is Dr. Kurtis Garbutt, email address: kurtis.garbutt@wrh.ox.ac.uk.

**Funding:** Mustafa Bahceci (Bahceci Health Group, Istanbul, Türkiye) has donated funds to the University of Oxford towards the study. Bethan Swift's DPhil funding, the Bahceci Scholarship, is donated by Mustafa Bahceci. Nilufer Rahmioglu has been crowdfunding for the project which raised a significant proportion of the funding necessary to conduct the study. Eastern Mediterranean University is funding the local data collection. The study received communication funding including telephones, tablets, and call minutes/3G support from Vodafone Mobile Operation Ltd. The project receives local support from Cyprus Women's Health Research Society (CoHERS), which is a registered charity in Northern Cyprus. Through CoHERS, the study received European Union Civic Space support to put together short films to promote the project and inform potential participants.

**Competing interests:** CMB declares that he is part of a scientific collaboration between Oxford University and Bayer Healthcare Ltd. for the purpose of drug target identification in endometriosis. He holds/has held research grants from Bayer Healthcare, MDNA Life Sciences and Roche Diagnostics and has in recent years been a consultant for Abbvie Inc., Roche Diagnostics, Myovant and ObsEva. KTZ declares that she has scientific collaborations outside the submitted work with Bayer Healthcare, MDNA Life Sciences, Roche Diagnostics Inc., and Volition Rx and is a Board member (Secretary) of the World Endometriosis Society, Research Advisory Board member of Wellbeing of Women, UK (research charity) and Chair of the Research Directions Working Group, World Endometriosis Society. NR declares that she is the founding president of Cyprus Women's Health Research Society in Northern Cyprus (CoHERS).

throughout the years at 88.7% and there was no significant association between breastfeeding initiation and the year of pregnancy, or demographic and maternal medical and pregnancy related variables. After full adjustment, women who gave birth after 2005 were 1.24 (95%CI; 1.06–1.45) times more likely to breastfeed for >12 weeks compared to women who gave birth before 1995. There was no association between C-section and breastfeeding prevalence or length.

## Conclusion

Prevalence of C-section in this population is much higher than WHO recommendations. Public awareness campaigns surrounding choice during pregnancy and change in legal framework to allow for midwife-led continuity models of birthing care should be implemented. Further research is required to understand the reasons and drivers behind this high rate.

## Introduction

Caesarean section (C-section) can be a life-saving intervention for both mother and child when medically necessary. In 2015 [1], the World Health Organization (WHO) released their statement on C-section stating that while maternal and neonatal mortality decreased as C-section rates increased towards 10%, C-section rates above 10% were not found to be associated with any reductions in maternal and newborn mortality and could have detrimental short- and long-term effects on the health of the mother and child [2, 3]. Despite WHO's guidance [4] on potential non-clinical interventions that could be used to help reduce unnecessary C-sections, recent trend analyses of 169 countries has shown that that between 2010–2018, 21% of women gave birth by C-section worldwide, with an average annual increase of 4% [5].

Cyprus is the third largest Mediterranean island with approximately 300,000 Turkish Cypriot and 700,000 Greek Cypriot residents (Box 1). There is a lack of population-level health data from the Turkish Cypriot population in Northern Cyprus which is an emerging region in Europe [6], and currently very limited data are available on C-section prevalence. C-section rates have been recently estimated as 55.3% in the Republic of Cyprus [5], but this excludes the northern part of the island where midwifery care is particularly limited; according to the Ministry of Health of Northern Cyprus [7], there were a total of 26 midwives employed in public hospitals across the region and approximately 3.53 nurses per 1,000 people. This low number is partly due to the fact that midwives are required to have an accredited midwifery degree, but there are no universities or institutions that offer this degree in Northern Cyprus [8]. In addition to this, the legal framework in Northern Cyprus means that deliveries can only be performed by clinicians in hospitals.

There is mixed evidence on whether C-section influences breastfeeding success. A systematic review and meta-analysis [9] showed a negative association between C-section and breastfeeding initiation but when breastfeeding was initiated, then C-section had no effect on the proportion of mothers continuing to do so up to 6 months. A study that took place in the Republic of Cyprus showed that although 84.3% of mothers initiated breastfeeding before being discharged, this figure decreased to 32.4% at 6 months, with mothers that gave birth vaginally being three times more likely to initiate breastfeeding (OR = 3.1; 95%CI 1.7,5.4) compared to those who gave birth by C-section [10].

## Box 1. Context of study

Located in the Eastern Mediterranean region, Cyprus is the third largest island in the Mediterranean. The island is divided into two parts; the internationally recognised Republic of Cyprus and the Turkish Republic of Northern Cyprus (Northern Cyprus), a *de facto* state only recognised by Türkiye. In 2004, the Republic of Cyprus became a member of the European Union and economically prospered. Conversely, Northern Cyprus has remained under economic sanctions for the past 40 years. There are approximately 300,000 people living in Northern Cyprus, the majority are Turkish Cypriot.

There are four potential pathways to accessing healthcare in Northern Cyprus [6]: 1) The public healthcare system–this is a heavily discounted service providing individuals with social security insurance, which is mandatory for everyone in the workforce, their partners and those under 18. Services in accident and emergency departments are free of charge for everyone. 2) The private healthcare system–although the proportions of individuals purchasing voluntary private health insurance has increased in recent years, it is not widespread so there are high out-of-pocket health care costs. 3) Public services in Türkiye–the Northern Cyprus government has a formal agreement with Türkiye whereby individuals can be sent to Türkiye free of charge for specialist healthcare if the required services are not available within the public sector. 4) Public services in the Republic of Cyprus–since Turkish Cypriots are eligible for citizenship from the Republic of Cyprus, some choose to cross the border and receive healthcare from the public services in the South. It is thought only a small percentage of Turkish Cypriots choose to do this. Similarly, patients can access gynaecology and obstetrics services in public healthcare for free or opt to access private healthcare via their private insurance coverage and at their own expenses. Due to the fragmented healthcare seeking behaviours, there is a lack of data on health needs and behaviours of the population of Northern Cyprus.

In Northern Cyprus, there have been various campaigns [11] surrounding breastfeeding awareness, but in the absence of any systematically collected data, there is no way of knowing what the prevalence of breastfeeding is, or whether these campaigns have made a difference to rates in the population. In addition, if the C-section rate is similarly high to the Republic of Cyprus, then this could have negative consequences on breastfeeding initiation and the subsequent length a woman chooses to breastfeed for.

Therefore, this study aimed to estimate the prevalence, trends and predictors of C-section and breastfeeding in Northern Cyprus and explore any associations between the two.

## Methods

### Study sample: The COHERE Initiative

This study uses data collected as part of the COHERE Initiative. The COHERE Initiative is a population based cross-sectional study that has recruited 7,646 consenting women between the ages of 18–55 in Northern Cyprus. The aim of COHERE Initiative is to determine the relative burden of women's health conditions and related co-morbidities in women living in Northern Cyprus and establish a women's health cohort for the future to investigate regional risk factors. At baseline, each participant completed a detailed culturally adapted questionnaire

on women's health expanding upon the standardised Endometriosis-Phenome-and-Biobanking-Harmonization-Project (EPHect) questions [12] to include questions on other women's health conditions and reproductive health as well. In particular, the questionnaire included questions on pregnancies history and breastfeeding per live birth. Data were collected through a combination of household (16%, (n = 1,208)) and workplace (84% (6,438)) face-to-face visits (93% (7,128)) as well as through online (7% (518)) recruitment methods [13], between January 2018 and February 2020. Women aged between 18 and 55 at recruitment, who were either citizens of Northern Cyprus or had been residing there for the past 5 years and were able to give informed consent were eligible to participate in the study. Women were recruited into the study from the 6 main districts in Northern Cyprus (Nicosia, Kyrenia, Famagusta, Morphou, Trikomo and Lefke) with recruitment targets being set using geographic population densities. We compared the demographics of the COHERE Initiative sample with projected 2019 population figures for Northern Cyprus obtained from the Northern Cyprus Statistics Institution (available on request from: http://www.stat.gov.ct.tr/). Our sample was broadly representative of these projected values with the main differences being seen for age and education in particular regions [14].

### Inclusion and exclusion criteria

The analysis was limited to first pregnancies only. Mothers with singleton live births between 1981 and 2017 who gave details on mode of delivery were included. We excluded all women under the age of 26 at the time of recruitment as the mean age of first pregnancy in our sample was 24.9, with the majority of women between 18–25 having not had any pregnancies yet. Women in this age group who had been pregnant were likely not representative of all women, so they were excluded to reduce the risk of biased estimates. We also excluded all women who had any missing data on any of the below variables examined.

### Outcome measures

C-section data was collected based on a woman's self-report to the COHERE survey question: 'If this pregnancy related to a birth, was the delivery vaginal or via Caesarean section?'. The C-section rate was defined as the percentage of all live singleton births born by C-section and we created a binary variable (vaginal/caesarean) to model the data. Data on ever breastfed and length breastfed for was based on the survey question: 'If this pregnancy resulted in a birth, for how long did you breastfeed?' where women were required to write the number of months they breastfed for. A response of 0 or non-response was taken as the woman not breastfeeding and we created a binary variable to analyse data on ever breastfeeding (yes/no). For women who breastfed, an additional binary variable was created to assess length of breastfeeding - ≤12 weeks (early cessation) or >12 weeks.

### Explanatory variables

Demographic variables included: age at pregnancy (<25, 25–29, 30–34, 35+), ethnicity (Turkish Cypriot, Turkish, Mixed/Other), education (Primary/middle school, high-school, post-secondary, undergraduate degree, postgraduate degree) and residence (city/village). Pregnancy-related variables included: pre-term birth (yes/no), fertility-treatment used (yes/no), and weight of the baby (underweight <2.5kg, normal weight, 2.5kg-4.0kg, overweight ≥4.0kg).

The primary exposures were year of pregnancy, which was calculated by summing a woman's birth year with the age at which she was pregnant. This was categorized into three groups: <1995, 1995–2005 and >2005. For the analysis of breastfeeding, the secondary exposure was mode of delivery (vaginal/caesarean).

## Statistical analysis

Descriptive statistics were used to describe the overall characteristics of the sample. To investigate the association between the exposures and outcomes, modified Poisson regression was used to estimate risk ratios (RRs) and 95% confidence intervals (CIs). For all models, univariable associations were estimated (model A) and then in a hierarchical fashion for potential confounders determined *a priori*: model B adjusted for sociodemographic factors; model C additionally adjusted for maternal medical and pregnancy-related factors. Confounders were determined *a priori* based on pre-existing hypotheses or evidence [15–18]. Variables were only included if they were associated with C-section or breastfeeding in univariable analysis. We excluded women who had any missing data on any of the variables included and performed a complete case analysis for all models.

Statistical analyses were carried out using Stata SE version 17.0 (StataCorp LP, College Station, Texas, USA) and figures were produced using R Studio.

## Ethics

The study was approved by the Oxford Tropical Research Ethics Committee (OxTREC) of the University of Oxford (OxTREC reference: 37–17). The study also received local ethics approval from the Eastern Mediterranean University Ethics Committee (ETK00-2017-0240). Verbal and written consent was obtained from all participants during recruitment.

## Results

### Study sample

Of the 7,646 women recruited into COHERE, 3,684 had a first singleton pregnancy, and 77.0% (n = 2,836) had complete data and were included in the study (Fig 1). The women included in the study and women excluded due to missing data were broadly similar except that women excluded tended to have lower educational attainment compared to those included in the study (S1 Table). Table 1 shows the characteristics of the study population overall as well as the prevalence of mode of delivery, whether they breastfed and if so, how long they breastfed for. Of the 2,836 births between 1981 and 2018, 55.3% (n = 1,568) were delivered by C-section. Women who gave birth after 2005 had a higher C-section rate (68.6%) than those who gave birth before 1995 (21.4%). The C-section rate was also higher in women who were older than 24 when they gave birth and had a higher educational achievement. Turkish Cypriot women had the highest C-section rate at 57.4%, compared to Turkish women (48.9%) and Other/Mixed women (54.3%). The C-section rate was also higher in women who had a preterm birth (69.4%) compared with a term birth (54.3%), and in those who had fertility treatment prior to their pregnancy (80.6%) compared with those who did not (54.0%).

### Trends in Caesarean section between 1981 and 2017

C-section rate increased dramatically from 11.1% in 1981 to 72.5% in 2017, with C-section births overtaking vaginal births in 1999 (Fig 2). Table 2 shows the crude RR of mode of birth by time, as well as after adjustment for sociodemographic and pregnancy related factors. Before any adjustment, women who gave birth between 1995–2005 (RR: 2.82; 95% CI 2.34 to 3.40) and those who gave birth after 2005 (RR: 3.20; 95% CI 2.67 to 3.84) had approximately a three-fold increased risk of having a Caesarean birth compared to women who gave birth before 1995. Adjustment for sociodemographic characteristics and pregnancy-related factors attenuated these RRs to 2.6 (95% CI 2.14 to 3.15) and 2.7 (95% CI 2.19 to 3.25), respectively.

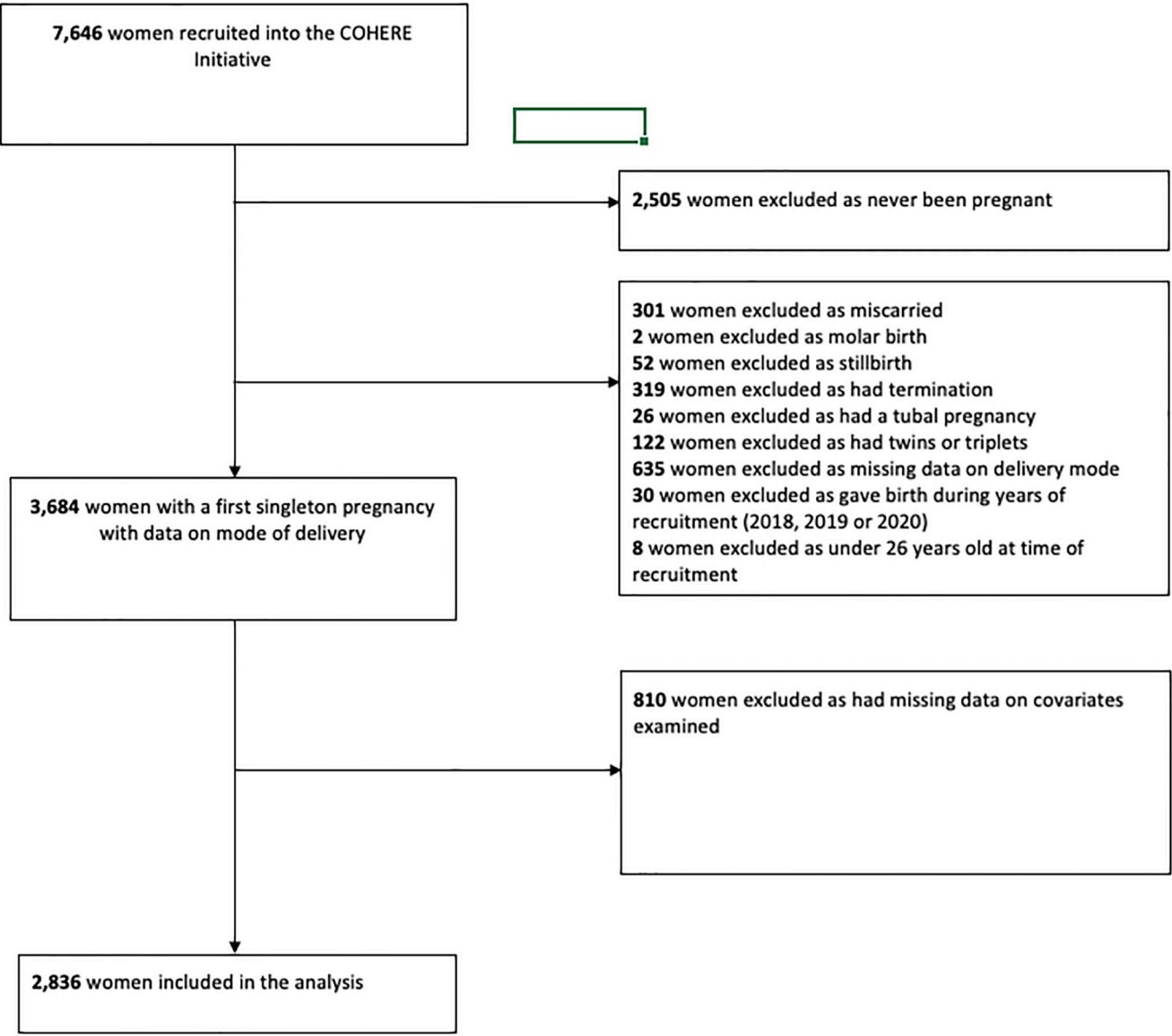

**Fig 1. Flow chart of women included in the analysis after applying the exclusion and inclusion criteria.**

After full adjustment, risk of giving birth by C-section increased with age with women who gave birth over the age of 35 having a 1.41 (95% CI 1.08 to 1.84) times greater risk compared to women who gave birth under 25 years of age. Women with a postgraduate degree had a 1.35 (95% CI 1.08 to 1.69) times greater risk of C-section compared to women with a primary/middle school degree after fully adjusting for sociodemographic and maternal medical and pregnancy-related factors.

### Trends in breastfeeding between 1981 and 2017

For all births, breastfeeding was initiated by 88.7% (n = 2,515) of women with prevalence rising from 85.4% in pregnancies before 1995 to 92.9% in pregnancies after 2005 (Table 1). Turkish Cypriot women had a slightly lower prevalence of breastfeeding (87.6% vs 91.2% in

**Table 1. Demographic characteristics and prevalence of caesarean section and breastfeeding in women who had a first pregnancy between 1981–2017.** Only includes women >25 years of age at time of recruitment.

| n (%)[a] | All | Caesarean section | | Breastfed^ | | Length breastfed for^* | |
|---|---|---|---|---|---|---|---|
| | (n = 2,836) | Yes (n = 1,568) | No (n = 1,268) | Yes (n = 2,515) | No (n = 321) | ≤12 weeks (n = 789) | >12 weeks (n = 1,726) |
| | | | | n (%) | | | |
| **Year of pregnancy** | | | | | | | |
| <1995 | 631 (22.3) | 135 (21.4) | 496 (78.6) | 539 (85.4) | 92 (14.6) | 211 (39.2) | 328 (60.9) |
| 1995–2005 | 961 (33.9) | 580 (60.4) | 381 (39.7) | 821 (85.4) | 140 (14.6) | 293 (35.7) | 528 (64.3) |
| >2005 | 1,244 (43.9) | 853 (68.6) | 391 (31.4) | 1,155 (92.9) | 89 (7.2) | 285 (24.7) | 870 (75.3) |
| **Age at pregnancy** | | | | | | | |
| <25 | 1,355 (47.8) | 560 (41.3) | 795 (58.7) | 1,181 (87.2) | 174 (12.8) | 425 (36.0) | 756 (64.0) |
| 25–29 | 1,051 (37.1) | 684 (65.1) | 367 (34.9) | 952 (90.6) | 99 (9.4) | 264 (27.7) | 688 (72.3) |
| 30–34 | 351 (12.4) | 256 (72.9) | 95 (27.1) | 316 (90.3) | 35 (10.0) | 79 (25.0) | 237 (75.0) |
| 35+ | 79 (2.8) | 68 (86.1) | 11 (13.9) | 66 (83.5) | 13 (16.5) | 21 (31.8) | 45 (68.2) |
| **Ethnicity** | | | | | | | |
| Turkish Cypriot | 2,037 (71.8) | 1,169 (57.4) | 868 (42.6) | 1,784 (87.6) | 253 (12.4) | 615 (34.5) | 1,340 (64.4) |
| Turkish | 648 (22.9) | 317 (48.9) | 331 (51.1) | 591 (91.2) | 57 (8.8) | 138 (23.4) | 496 (75.6) |
| Mixed/Other | 151 (5.3) | 82 (54.3) | 69 (45.7) | 140 (92.7) | 11 (7.3) | 36 (25.7) | 112 (71.8) |
| **Education** | | | | | | | |
| Primary/Middle school | 448 (15.8) | 152 (33.9) | 296 (66.1) | 400 (89.3) | 48 (10.7) | 98 (24.5) | 302 (75.5) |
| High school/Post-secondary | 1,005 (35.4) | 524 (52.1) | 481 (47.9) | 846 (84.2) | 159 (15.8) | 335 (39.6) | 511 (60.4) |
| Undergraduate degree | 946 (33.4) | 590 (62.4) | 356 (37.6) | 855 (90.4) | 91 (9.6) | 260 (30.4) | 595 (69.6) |
| Postgraduate degree | 437 (15.4) | 302 (69.1) | 135 (30.9) | 414 (94.7) | 23 (5.3) | 96 (23.2) | 318 (76.8) |
| **Residence type** | | | | | | | |
| Village | 1,567 (55.3) | 866 (55.3) | 701 (44.7) | 1,393 (88.9) | 174 (11.1) | 433 (31.1) | 960 (68.9) |
| City | 1,269 (44.8) | 702 (55.3) | 567 (44.7) | 1,122 (88.4) | 147 (11.6) | 356 (31.7) | 766 (68.3) |
| **Pre-term birth** | | | | | | | |
| No | 2,650 (93.4) | 1,439 (54.3) | 1,211 (45.7) | 2,363 (89.2) | 287 (10.8) | 729 (30.9) | 1,634 (69.2) |
| Yes | 186 (6.6) | 129 (69.4) | 57 (30.7) | 152 (81.7) | 34 (18.3) | 60 (39.5) | 92 (60.5) |
| **IVF treatment** | | | | | | | |
| No | 2,702 (95.3) | 1,460 (54.0) | 1,242 (46.0) | 2,397 (88.7) | 305 (11.3) | 748 (31.2) | 1,649 (68.8) |
| Yes | 134 (4.7) | 108 (80.6) | 26 (19.4) | 118 (86.1) | 16 (11.9) | 41 (34.8) | 77 (65.3) |
| **Baby weight (kg)** | | | | | | | |
| Normal (≥2.5kg—<4.0kg) | 2,409 (84.9) | 1,323 (54.9) | 1,086 (45.1) | 2,149 (89.2) | 260 (10.8) | 650 (30.3) | 1,499 (69.8) |
| Under (<2.5kg) | 198 (7.0) | 115 (58.1) | 83 (41.9) | 165 (83.3) | 33 (16.7) | 71 (43.0) | 94 (57.0) |
| Over (≥4.0kg) | 229 (8.1) | 130 (56.8) | 99 (43.2) | 201 (87.8) | 28 (12.2) | 68 (33.8) | 133 (66.2) |

[a]Percentages in the first column (All) are column percentages, the rest are row percentages

^Includes all first births regardless of whether it was c-section or vaginal

*Only includes women who breastfed

Turkish women and 91.2% in Other/Mixed women) and women with a postgraduate degree had the highest breastfeeding prevalence (94.7%). Prevalence of a post-graduate degree in Turkish women was 7.1% compared to 17.8% of Turkish Cypriot women and Turkish women had lower levels of employment (68.2%) compared to Turkish Cypriot (88.0%) and Mixed/Other (88.7%) women, p<0.001 (data not shown). Breastfeeding prevalence was slightly lower in mothers who had pre-term births (81.7% vs 89.2%) and those who used fertility treatments (86.1% vs 88.7%). Of the women who breastfed, 68.6% (n = 1,726) did so for over 12 weeks and the proportion of women doing so rose from

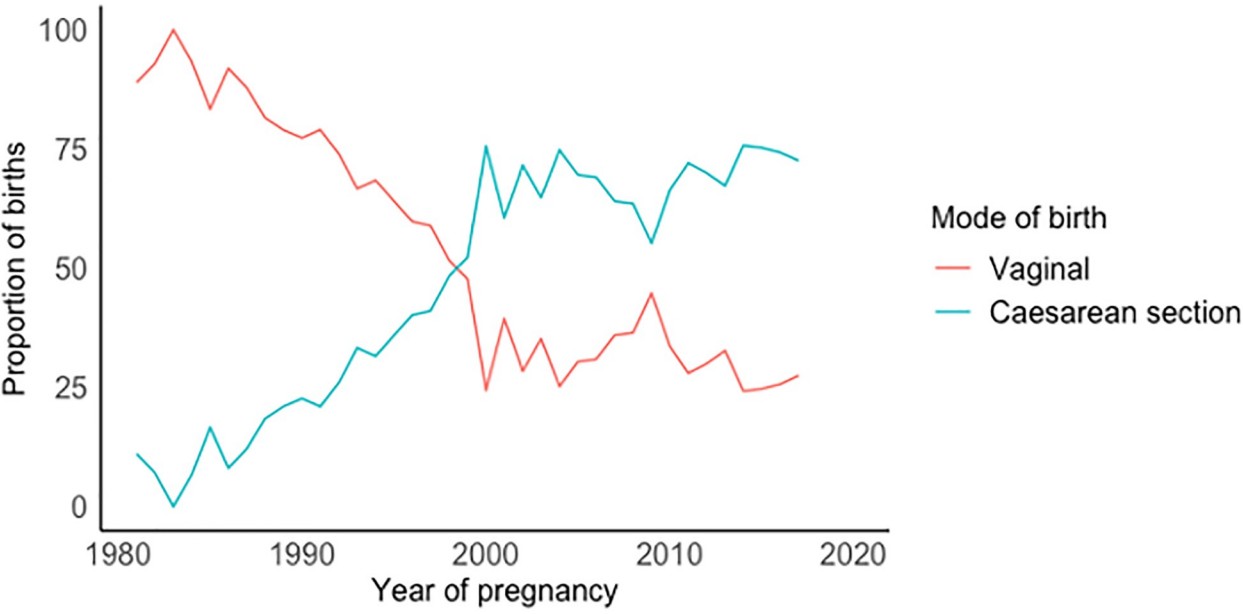

**Fig 2. Proportion of women whose first birth was vaginal (red) or caesarean (blue).**

60.9% of births before 1995 to 75.3% of births after 2005. Women aged <25 at pregnancy had the lowest rate of breastfeeding for >12 weeks (64.0%) as did Turkish Cypriot women (64.4% vs 75.6% in Turkish and 71.8% in Other/Mixed). Breastfeeding for the longer duration was lower after pre-term birth (60.5% vs 69.2%) and after births from women who had used fertility treatments (65.3% vs 68.8%). Rates were lowest in babies born underweight (57.0%).

Prevalence of ever breastfeeding has remained high between 1981 and 2017, with prevalence decreasing from 100% in 1981 to 94% in 2016, before dropping to 82% in 2017 (Fig 3a). Prevalence was lowest at 72% in 1992. Of the women who breastfed, 12.2% breastfed for 1 month or less, 57.3% breastfed for 6 or more months and 28.9% breastfed for 12 or more months. Table 3 shows the crude and adjusted RR (aRR) of ever breastfed by year of pregnancy. After adjustment for both sociodemographic and maternal medical and pregnancy-related factors, women with pregnancies after 2005 had a 10% higher prevalence (aRR: 1.10; 95% CI 0.97 to 1.24) of breastfeeding compared to women with pregnancies before 1995, but this was not statistically significant. None of the sociodemographic or maternal medical and pregnancy-related factors examined had a significant effect on breastfeeding initiation.

Fig 3b depicts the proportion of time women breastfed for (≤12/>12 weeks). The proportion of women breastfeeding for over 12 weeks increased from 44.4% in 1981 to 73.3% in 2019, with the proportion of women breastfeeding for ≤12 weeks decreasing from 55.6% in 1981 to 24.7% in 2017; women who gave birth after 2005 were 1.2 times more likely (RR: 1.24; 95% CI 1.09 to 1.41) to breastfeed for >12 weeks compared to women who gave birth before 1995 (Table 3) and after full adjustment, the effect estimate did not attenuate. Compared to Turkish Cypriot women, Turkish women were more likely to breastfeed for longer and women with high school/post-secondary education were less likely to breastfeed compared to women with primary/middle school education.

**Table 2. Relative risk of caesarean section for first pregnancy in relation to year of pregnancy adjusted for sociodemographics (Model 1) and additionally for pregnancy-related factors (Model 2).** Only includes women >25 years of age at time of recruitment.

| | Crude (complete case n = 2,836) | | Model 1 (complete case n = 2,836)[a] | | Model 2 (complete case n = 2,836)[a] | |
|---|---|---|---|---|---|---|
| | RR | (95% CI) | RR | (95% CI) | RR | (95% CI) |
| **Year of pregnancy** | | | | | | |
| <1995 | 1 | - | 1 | - | 1 | - |
| 1995–2005 | 2.82*** | (2.34, 3.40) | 2.60*** | (2.15, 3.15) | 2.60*** | (2.14, 3.15) |
| >2005 | 3.20*** | (2.67, 3.84) | 2.65*** | (2.18, 3.24) | 2.67*** | (2.19, 3.25) |
| **Age at pregnancy** | | | | | | |
| <25 | - | - | 1 | - | 1 | - |
| 25–29 | - | - | 1.19** | (1.05, 1.35) | 1.19** | (1.04, 1.35) |
| 30–34 | - | - | 1.26** | (1.06, 1.49) | 1.24* | (1.05, 1.47) |
| 35+ | - | - | 1.47** | (1.13, 1.91) | 1.41* | (1.08, 1.84) |
| **Ethnicity** | | | | | | |
| Turkish Cypriot | - | - | 1 | - | 1 | - |
| Turkish | - | - | 0.95 | (0.83, 1.09) | 0.95 | (0.83, 1.09) |
| Mixed/Other | - | - | 0.89 | (0.71, 1.11) | 0.89 | (0.71, 1.11) |
| **Education** | | | | | | |
| Primary/Middle school | - | - | 1 | - | 1 | - |
| High school/Post-secondary | - | - | 1.33** | (1.10, 1.61) | 1.32** | (1.09, 1.60) |
| Undergraduate degree | - | - | 1.34** | (1.09, 1.63) | 1.33** | (1.09, 1.63) |
| Postgraduate degree | - | - | 1.36** | (1.09, 1.70) | 1.35** | (1.08, 1.69) |
| **Residence type** | | | | | | |
| Village | - | - | 1 | - | 1 | - |
| City | - | - | 0.98 | (0.89, 1.09) | 0.99 | (0.89, 1.09) |
| **Pre-term birth** | | | | | | |
| No | - | - | - | - | 1 | - |
| Yes | - | - | - | - | 1.15 | (0.96, 1.39) |
| **IVF treatment** | | | | | | |
| No | - | - | - | - | 1 | - |
| Yes | - | - | - | - | 1.22* | (1.00, 1.50) |
| **Baby weight (kg)** | | | | | | |
| Normal (≥2.5kg—<4.0kg) | - | - | - | - | 1 | - |
| Under (<2.5kg) | - | - | - | - | 1.07 | (0.88, 1.31) |
| Over (≥4.0kg) | - | - | - | - | 1.21 | (1.00, 1.45) |

***p<0.001,

**p<0.01,

*p<0.05

[a]Does not include variables that were not associated with c-section or breastfeeding in univariable analysis (BMI, gestational diabetes, gender of baby, hypertension, hyperemesis gravidarum)

## Relationship between Caesarean section and breastfeeding initiation and length

The proportion of women who breastfed was 87.8% in those who had a Caesarean section and 89.8% in those who had a vaginal birth. C-section had no association with whether a woman reported to ever breastfeed or not; after adjusting for sociodemographic and maternal medical and pregnancy-related factors, there was no significant relationship between mode of delivery and ever breastfeeding (Table 4). There was also no significant relationship between mode of

(a)

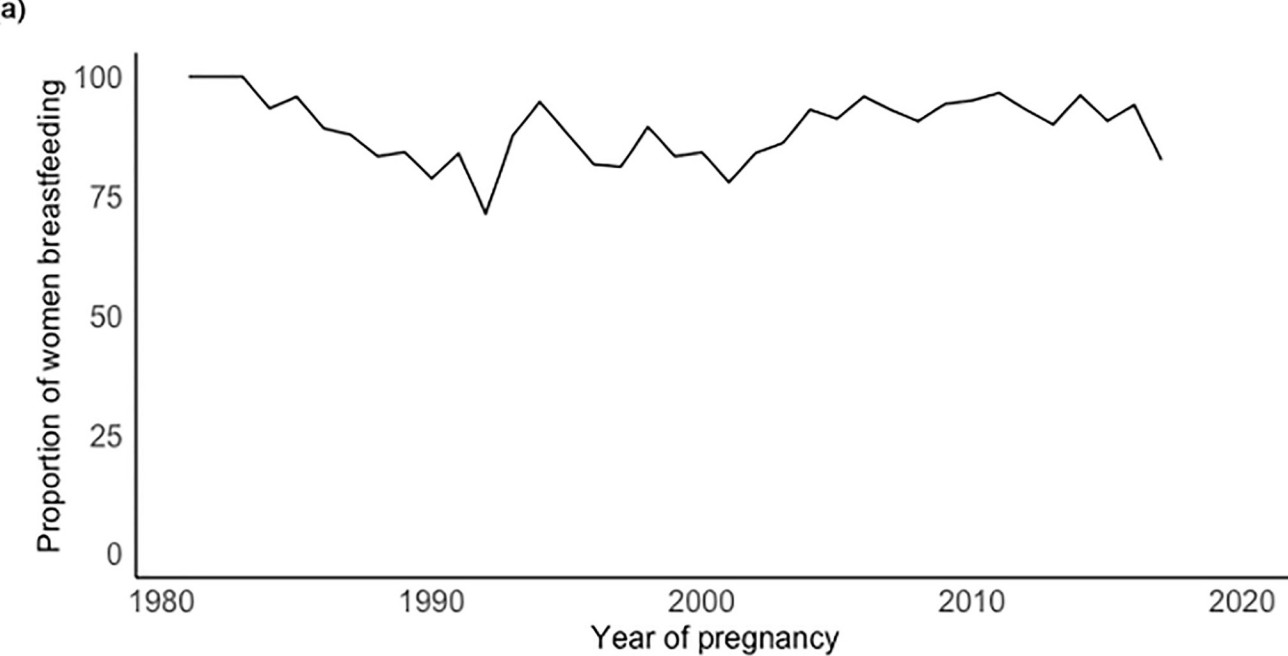

(b)

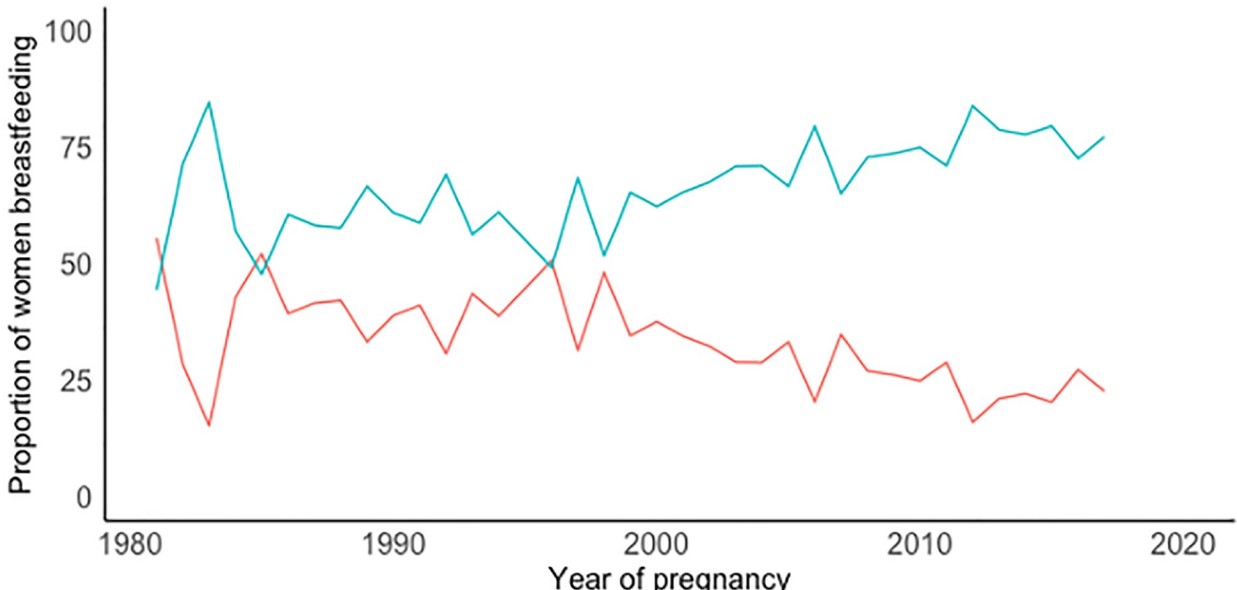

**Fig 3. a) Proportion of women who ever initiated breastfeeding during their first pregnancy by year of pregnancy and b) proportion of women who breastfed for ≤)12 weeks and ii) >12 weeks.** Denominator only includes women who ever breastfed.

**Table 3. Relative risk of breastfeeding (yes/no) and length of breastfeeding (≤12/>12 weeks) for first pregnancy in relation to year of pregnancy adjusted for socio-demographics (Model 1) and additionally for pregnancy-related factors (Model 2).** Only includes women >25 years of age at time of recruitment.

| | Breastfeeding (yes/no) | | | | | | Length of breastfeeding (≤12/>12 weeks)^ | | | | | |
| | Crude (complete case n = 2,836) | | Model 1 (complete case n = 2,836)[a] | | Model 2 (complete case n = 2,836)[a] | | Crude (complete case n = 2,515) | | Model 1 (complete case n = 2,515)[a] | | Model 2 (complete case n = 2,515)[a] | |
| | RR | (95% CI) | RR | (95% CI) | RR | (95% CI) | RR | (95% CI) | RR | (95% CI) | RR | (95% CI) |
|---|---|---|---|---|---|---|---|---|---|---|---|---|
| **Year of pregnancy** | | | | | | | | | | | | |
| <1995 | 1 | - | 1 | - | 1 | - | 1 | - | 1 | - | 1 | - |
| 1995–2005 | 1.00 | (0.90, 1.11) | 1.00 | (0.89, 1.12) | 1.01 | (0.90, 1.14) | 1.06 | (0.92, 1.21) | 1.04 | (0.90, 1.20) | 1.08 | (0.93, 1.25) |
| >2005 | 1.09 | (1.98, 1.20) | 1.08 | (0.96, 1.22) | 1.10 | (0.97, 1.24) | 1.24** | (1.09, 1.41) | 1.19* | (1.03, 1.38) | 1.24** | (1.06, 1.45) |
| **Age at pregnancy** | | | | | | | | | | | | |
| <25 | - | - | 1 | - | 1 | - | - | - | 1 | - | 1 | - |
| 25–29 | - | - | 0.99 | (0.90, 1.09) | 0.99 | (0.90, 1.10) | - | - | 1.07 | (0.95, 1.21) | 1.08 | (0.95, 1.22) |
| 30–34 | - | - | 0.96 | (0.83, 1.10) | 0.96 | (0.83, 1.11) | - | - | 1.06 | (0.90, 1.25) | 1.08 | (0.91, 1.28) |
| 35+ | - | - | 0.90 | (0.69, 1.16) | 0.91 | (0.70, 1.18) | - | - | 0.97 | (0.71, 1.33) | 1.02 | (0.74, 1.41) |
| **Ethnicity** | | | | | | | | | | | | |
| Turkish Cypriot | - | - | 1 | - | 1 | - | - | - | 1 | - | 1 | - |
| Turkish | - | - | 1.05 | (0.94, 1.16) | 1.04 | (0.94, 1.15) | - | - | 1.15* | (1.02, 1.30) | 1.14* | (1.01, 1.29) |
| Mixed/Other | - | - | 1.06 | (0.89, 1.25) | 1.05 | (0.88, 1.25) | - | - | 1.12 | (0.91, 1.37) | 1.11 | (0.91, 1.36) |
| **Education** | | | | | | | | | | | | |
| Primary/Middle school | - | - | 1 | - | 1 | - | - | - | 1 | - | 1 | - |
| High school/Post-secondary | - | - | 0.96 | (0.84, 1.09) | 0.96 | (0.85, 1.10) | - | - | 0.83* | (0.71, 0.96) | 0.83* | (0.71, 0.97) |
| Undergraduate degree | - | - | 1.02 | (0.88, 1.17) | 1.02 | (0.89, 1.18) | - | - | 0.90 | (0.76, 1.06) | 0.91 | (0.77, 1.07) |
| Postgraduate degree | - | - | 1.07 | (0.91, 1.25) | 1.07 | (0.91, 1.26) | - | - | 0.97 | (0.80, 1.17) | 0.98 | (0.81, 1.19) |
| **Residence type** | | | | | | | | | | | | |
| Village | - | - | 1 | - | 1 | - | - | - | 1 | - | 1 | - |
| City | - | - | 0.99 | (0.92, 1.08) | 0.99 | (0.92, 1.08) | - | - | 0.99 | (0.90, 1.09) | 0.99 | (0.90, 1.09) |
| **C-section** | | | | | | | | | | | | |
| No | - | - | - | - | 1 | - | - | - | - | - | 1 | - |
| Yes | - | - | - | - | 0.96 | (0.88, 1.04) | - | - | - | - | 0.91 | (0.82, 1.00) |
| **Pre-term birth** | | | | | | | | | | | | |
| No | - | - | - | - | 1 | - | - | - | - | - | 1 | - |
| Yes | - | - | - | - | 0.92 | (0.78, 1.09) | - | - | - | - | 0.91 | (0.73, 1.13) |
| **IVF treatment** | | | | | | | | | | | | |
| No | - | - | - | - | 1 | - | - | - | - | - | 1 | - |
| Yes | - | - | - | - | 1.00 | (0.83, 1.21) | - | - | - | - | 0.92 | (0.73, 1.16) |
| **Baby weight (kg)** | | | | | | | | | | | | |
| Normal (≥2.5kg—<4.0kg) | - | - | - | - | 1 | - | - | - | - | - | 1 | - |
| Under (<2.5kg) | - | - | - | - | 0.96 | (0.81, 1.13) | - | - | - | - | 0.84 | (0.68, 1.04) |
| Over (≥4.0kg) | - | - | - | - | 1.00 | (0.86, 1.16) | - | - | - | - | 0.98 | (0.82, 1.18) |

***p<0.001,

**p<0.01,

*p<0.05

^Denominator only includes women who ever breastfed

[a]Does not include variables that were not associated with c-section or breastfeeding in univariable analysis (BMI, gestational diabetes, gender of baby, hypertension, hyperemesis gravidarum)

**Table 4. Relative risk of breastfeeding (yes/no) and length of breastfeeding (≤12/>12 weeks) for first pregnancy in relation to mode of birth adjusted for sociode-mographics (Model 1) and additionally for pregnancy-related factors (Model 2).** Only includes women >25 years of age at time of recruitment.

| | Breastfeeding (yes/no) | | | | | | Length of breastfeeding (≤12/>12 weeks)^ | | | | | |
| | Crude (complete case n = 2,836) | | Model 1 (complete case n = 2,836)[a] | | Model 2 (complete case n = 2,836)[a] | | Crude (complete case n = 2,515) | | Model 1 (complete case n = 2,515)[a] | | Model 2 (complete case n = 2,515)[a] | |
| | RR | (95% CI) | RR | (95% CI) | RR | (95% CI) | RR | (95% CI) | RR | (95% CI) | RR | (95% CI) |
|---|---|---|---|---|---|---|---|---|---|---|---|---|
| **C-section** | | | | | | | | | | | | |
| No | 1 | - | 1 | - | 1 | - | 1 | - | 1 | - | 1 | - |
| Yes | 0.98 | (0.90, 1.06) | 0.96 | (0.88, 1.04) | 0.96 | (0.88, 1.04) | 0.96 | (0.87, 1.05) | 0.90* | (0.81, 0.99) | 0.91 | (0.82, 1.00) |
| **Year of pregnancy** | | | | | | | | | | | | |
| <1995 | - | - | 1 | - | 1 | - | - | - | 1 | - | 1 | - |
| 1995–2005 | - | - | 1.01 | (0.90, 1.14) | 1.01 | (0.90, 1.14) | - | - | 1.08 | (0.93, 1.25) | 1.08 | (0.93, 1.25) |
| >2005 | - | - | 1.10 | (0.97, 1.24) | 1.10 | (0.97, 1.24) | - | - | 1.24** | (1.06, 1.44) | 1.24** | (1.06, 1.45) |
| **Age at pregnancy** | | | | | | | | | | | | |
| <25 | - | - | 1 | - | 1 | - | - | - | 1 | - | 1 | - |
| 25–29 | - | - | 0.99 | (0.90, 1.10) | 0.99 | (0.90, 1.10) | - | - | 1.08 | (0.96, 1.22) | 1.08 | (0.95, 1.22) |
| 30–34 | - | - | 0.96 | (0.84, 1.11) | 0.96 | (0.83, 1.11) | - | - | 1.08 | (0.91, 1.28) | 1.08 | (0.91, 1.28) |
| 35+ | - | - | 0.91 | (1.70, 1.17) | 0.91 | (0.70, 1.18) | - | - | 1.01 | (0.73, 1.38) | 1.02 | (0.74, 1.41) |
| **Ethnicity** | | | | | | | | | | | | |
| Turkish Cypriot | - | - | 1 | - | 1 | - | - | - | 1 | - | 1 | - |
| Turkish | - | - | 1.05 | (0.94, 1.16) | 1.04 | (0.94, 1.16) | - | - | 1.15* | (1.02, 1.30) | 1.14* | (1.01, 1.29) |
| Mixed/Other | - | - | 1.05 | (0.89, 1.25) | 1.05 | (0.88, 1.25) | - | - | 1.11 | (0.91, 1.36) | 1.11 | (0.91, 1.36) |
| **Education** | | | | | | | | | | | | |
| Primary/Middle school | - | - | 1 | - | 1 | - | - | - | 1 | - | 1 | - |
| High school/Post-secondary | - | - | 0.96 | (0.85, 1.10) | 0.96 | (0.85, 1.10) | - | - | 0.84* | (0.72, 0.98) | 0.83* | (0.71, 0.97) |
| Undergraduate degree | - | - | 1.02 | (0.89, 1.17) | 1.02 | (0.89, 1.18) | - | - | 0.91 | (0.77, 1.07) | 0.91 | (0.77, 1.07) |
| Postgraduate degree | - | - | 1.07 | (0.91, 1.26) | 1.07 | (0.91, 1.26) | - | - | 0.99 | (0.82, 1.19) | 0.98 | (0.81, 1.19) |
| **Residence type** | | | | | | | | | | | | |
| Village | - | - | 1 | - | 1 | - | - | - | 1 | - | 1 | - |
| City | - | - | 0.99 | (0.92, 1.08) | 0.99 | (0.92, 1.08) | - | - | 0.99 | (0.90, 1.09) | 0.99 | (0.90, 1.09) |
| **Pre-term birth** | | | | | | | | | | | | |
| No | - | - | - | - | 1 | - | - | - | - | - | 1 | - |
| Yes | - | - | - | - | 0.92 | (0.78, 1.09) | - | - | - | - | 0.91 | (0.73, 1.13) |
| **IVF treatment** | | | | | | | | | | | | |
| No | - | - | - | - | 1 | - | - | - | - | - | 1 | - |
| Yes | - | - | - | - | 1.00 | (0.83, 1.21) | - | - | - | - | 0.92 | (0.73, 1.16) |
| **Baby weight (kg)** | | | | | | | | | | | | |
| Normal (≥2.5kg—<4.0kg) | - | - | - | - | 1 | - | - | - | - | - | 1 | - |
| Under (<2.5kg) | - | - | - | - | 0.96 | (0.81, 1.13) | - | - | - | - | 0.84 | (0.68, 1.04) |
| Over (≥4.0kg) | - | - | - | - | 1.00 | (0.86, 1.16) | - | - | - | - | 0.98 | (0.82, 1.18) |

***p<0.001,

**p<0.01,

*p<0.05

^Only includes women who ever breastfed

[a]Does not include variables that were not associated with c-section or breastfeeding in univariable analysis (BMI, gestational diabetes, gender of baby, hypertension, hyperemesis gravidarum)

delivery and length of breastfeeding in women who initiated breastfeeding after adjusting for the same factors.

## Discussion

Using data from the nationally representative COHERE Initiative, we have estimated the trends in the prevalence of C-section and breastfeeding between 1981–2017 and examined a range of sociodemographic and maternal medical and pregnancy-related risk factors associated with these as well as the relationship between C-section and breastfeeding. C-section rate amongst first pregnancies has dramatically increased from 1981–2017 and year of pregnancy was the strongest risk factor associated with an increased risk of C-section. Other strong risk factors were older age at pregnancy and higher education level. None of the maternal medical and pregnancy-related characteristics examined here were associated with C-section apart from fertility treatment, which showed a small increase in risk. Prevalence of breastfeeding has remained high between 1981–2017 and we did not see any statistically significant relationships between C-section and ever breastfeeding. In women who initiated breastfeeding, more women appear to be breastfeeding for a longer period of time (>12 weeks) and indeed, year of pregnancy was the strongest risk factor for doing so. Ethnicity and education had small significant associations with length of breastfeeding, but none of the maternal medical or pregnancy-related factors were statistically significant. We saw no significant relationships between Caesarean section and both breastfeeding outcomes.

Though this is the first-time the C-section rate has been described in Northern Cyprus comprehensively, the Northern Cyprus Ministry of Health estimated the C-section rate to be 47.6% [19] in 2017 in one comprehensive state hospital. Another small study that took place between 2015–2016 estimated C-section rate to be 87% [20] in one private hospital, but in this study, birth mode was not the primary outcome or focus of the study. In the Republic of Cyprus in 2017, it was estimated that the rate C-section was 55.3% [21] for all live births. According to our study, the C-section rate in Northern Cyprus in 2017 was considerably higher at 72.5%. We found that C-sections were higher in women who were older, more educated and in those who had used fertility treatments, which is not freely available. This may highlight existing inequalities in the region, and we can hypothesise that women who are older and more educated are more affluent and therefore have more access to information and current societal views on their birth options which could potentially shift their birth preference to C-section over vaginal births. In addition, women who are older may have more difficulties conceiving compared to younger women and use fertility treatments which can increase the likelihood of experiencing complexities during pregnancy and subsequently the need to have a C-section. However, these factors do not account for the large, overall increase in C-section births over the years, which suggests the processes of pregnancy and birth are highly medicalised. Furthermore, previous studies [22, 23], have shown previous C-section to be a driving factor of subsequent C-sections but as we limited our analysis to first pregnancies, the high rate shown here is even more striking.

Though we did not collect qualitative information about reasons for C-section as part of our study, extensive work carried out in the Republic of Cyprus revealed some of the most common reasons given by mothers were lack of knowledge and discussion on the benefits and risks, fear of the unknown, increases in fertility treatments and unethical behaviours of some obstetricians surrounding financial gain [24, 25]. In Northern Cyprus, obstetricians are also paid higher for a C-section birth compared to a vaginal birth. Similarly, there may be factors that are related to the health systems, providers and facilities that influence C-section rates; research in other settings has found that some providers prefer to perform C-sections over

vaginal births due to uncertainties around birth timings, fears of litigation, personal convenience and too few labour rooms [26, 27]. Since clinicians are the only health-care providers who are legally allowed to deliver babies in Northern Cyprus, all births must take place in hospitals and there is lack of midwives, it can be hypothesised that the high C-section rates are partly due to clinicians needing to schedule deliveries as they are solely responsible for safe births. The midwife-led continuity model of care is where care is provided from the same midwife or team of midwives during pregnancy, birth, and the early parenting period. A Cochrane review [28] of 15 studies suggested that women receiving midwife-led continuity models of care are less likely to experience interventions during their pregnancies and more likely to be satisfied with their care compared to women that receive other models of care. Research [29] has also shown that midwifery-led care can decrease C-section rates. However, there are no institutions in Northern Cyprus that offer a midwifery degree, which makes the situation more challenging.

Projections on global C-section rates estimate [5] that by 2030, 28.5% of women worldwide will give birth by C-section, ranging from 7.1% in sub-Saharan Africa, to 63.4% in Eastern Asia. The rate we have observed here in Northern Cyprus is particularly worrying as in 2017, the estimated rate has already passed the highest estimate for 2030.

The overall rate of mothers ever initiating breastfeeding between 1981–2017 was estimated at 88.7%, and although direct comparison is not possible due to differing methodologies and time periods studied, this is similar to rates in the Republic of Cyprus (84.3%) [10], higher than some countries such as England (72.0%) [30] and Malta (64.4%) but lower than others such as Türkiye (97.4%) [31] and Italy (96.6%) [32]. Prevalence of breastfeeding at 6 months was estimated at 57.3%, which is lower than Türkiye (87%) [33], higher than the Republic of Cyprus (32.4%) [10], England (34%) [33] and Italy (5.5%) [34], though we did not have information on whether this was exclusive breastfeeding or supplementary. Between 1981 and 2017, the proportion of mothers choosing to breastfeed for >12 weeks compared to ≤12 weeks significantly increased, with Turkish women being more likely to breastfeed for longer. Recently, there have been efforts by the Health Ministry in Türkiye to promote breastfeeding and the average duration of breastfeeding is thought to be around 17 months [35].

Importantly, we saw no relationship between C-section and breastfeeding initiation or length of breastfeeding. Research in Türkiye [31] showed the most important predictor of early initiation of breastfeeding was vaginal delivery and a study in Republic of Cyprus [10] showed that delivery by C-section was the strongest negative determinant of breastfeeding during the first 48 hours. A study in Canada [36] also saw that planned C-section is associated with early breastfeeding cessation, which is again not in line with the results from our study.

The high rate of C-section in this population may also lead to loss of procedural skills for obstetricians/gynaecologists. If healthcare providers do not frequently carry out vaginal births, then this may impact their preparedness, capability, and capacity to do so. The Term Breech Trial [37] was an international, multicentre trial that investigated the benefits of planned C-section over vaginal birth for women with a breech presentation. The Trial concluded that planned C-section would significantly reduce the incidence of neonatal morbidity and foetal death, which subsequently led to national organisations [36] issuing statements in favour of planned C-section for women with foetuses in a breech presentation. It was later revealed that the Term Breech Trial suffered from methodological flaws and issues with the analysis [38]. Unfortunately, this has led to a loss of skills in healthcare providers which means that women with a breech presentation at term who wish to have a vaginal delivery or are already in the second stages of labour are not able to do so, the latter of which may lead to foetal injury in an emergency.

## Strengths and limitations

To our knowledge, this is the first comprehensive study that has explored C-section and breastfeeding rates in Northern Cyprus. We have used data from the COHERE Initiative, which consists of a sample that is broadly representative of women living in Northern Cyprus. It provides critical evidence on an increasing worldwide trend and explores several different outcomes.

There are limitations to this study. Due to the cross-sectional study design causality cannot be inferred and as the study asks questions about past pregnancies, the results may suffer from recall bias. Information on sociodemographic factors was collected at time of recruitment rather than time of pregnancy so some e.g., residency, may have changed. The women included in the analysis may not be representative of all women who gave birth between 1981–2017. We did not have information on important factors such as whether C-sections were planned or whether breastfeeding was exclusive and therefore there may be unmeasured confounding present in our analyses. We also did not have data on other risk factors for C-section such as BMI at time of pregnancy. However, this research does serve as a starting point for more targeted follow-up studies and provides a snapshot of what is happening at a population level.

## Implications for research and practice

Further work could involve public awareness campaigns and advocacy activities to change the legal framework to allow midwives to deliver babies and educate and inform women and their families about the risks associated with C-section. Long term, this would require adaptation of the midwifery standards in the EU to Northern Cyprus and specialist training for both midwives and clinicians, including the ability to be able to obtain a midwifery degree on the island. In terms of public awareness campaigns, similar initiatives in Italy [39] that aimed to generate awareness about the increase and overuse of C-section while engaging with parliament have been successful. Similarly, the Republic of Cyprus created Birth Forward [40], which is a successful initiative that provides advocacy, support and education for families and professionals in regards to the stages of planning, creating and growing a family. A designated committee within the Ministry of Health who could be responsible for driving change, as in the Republic of Cyprus, would also be a positive step forwards [25].

## Conclusions

For the first time we have described the C-section rate in Northern Cyprus and demonstrated that it has increased dramatically, and it is much higher than the recommended levels. Breastfeeding initiation was high and there has been an increase in length of breastfeeding over the years. There was no relationship between Caesarean section and breastfeeding initiation or length. Advocacy activities to change the legal framework to allow midwives to deliver babies as well as public awareness campaigns and health interventions to inform women of their choice during pregnancy should be considered.

## Supporting information

**S1 Table. Demographic characteristics and comparison between women included and excluded from the study due to missing data.**
(DOCX)

## Acknowledgments

We would like to thank all the women who volunteered to take part in our study and our research assistants from the Eastern Mediterranean University who conducted the baseline recruitment in the field. Namely, Ecem Fidan, Kamil Ipciler, Asya Koparan, Cise Mis, Elif Okur, Zisan Pekri, Pembe Savas, Melis Tarhan, Ebru Yasti Gokcen Kofali, Gizem Gurpinar, Sumeyye Istanbul, Dilara Altan, and the data entry team. We are grateful to the Cyprus Women's Health Research Society (CoHERS) civil society members for their support of the project in spreading the word and reaching the women in Northern Cyprus.

## Author Contributions

**Conceptualization:** Bethan Swift, Bahar Taneri, Ilgin Cagnan, Krina T. Zondervan, Maria A. Quigley, Nilufer Rahmioglu.

**Data curation:** Bethan Swift.

**Formal analysis:** Bethan Swift, Maria A. Quigley.

**Funding acquisition:** Nilufer Rahmioglu.

**Investigation:** Bethan Swift, Maria A. Quigley, Nilufer Rahmioglu.

**Methodology:** Bethan Swift, Maria A. Quigley, Nilufer Rahmioglu.

**Resources:** Nilufer Rahmioglu.

**Supervision:** Bahar Taneri, Ilgin Cagnan, Christian M. Becker, Krina T. Zondervan, Maria A. Quigley, Nilufer Rahmioglu.

**Writing – original draft:** Bethan Swift.

**Writing – review & editing:** Bethan Swift, Bahar Taneri, Ilgin Cagnan, Christian M. Becker, Krina T. Zondervan, Maria A. Quigley, Nilufer Rahmioglu.

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
