## [Decision Letter · Decision Letter 0]

6 Jun 2023

Predictors and trends of Caesarean section and breastfeeding in the Eastern Mediterranean region: data from the cross-sectional Cyprus Women’s Health Research (COHERE) Initiative

PONE-D-22-23791

Dear Dr. Rahmioglu,

We’re pleased to inform you that your manuscript has been judged scientifically suitable for publication and will be formally accepted for publication once it meets all outstanding technical requirements.

Kind regards,

Mona Nabulsi, MD, MS

Academic Editor

PLOS ONE

Additional journal requirements:

Please provide additional details regarding participant consent. In the ethics statement in the Methods and online submission information, please ensure that you have specified what type you obtained (for instance, written or verbal, and if verbal, how it was documented and witnessed). If your study included minors, state whether you obtained consent from parents or guardians. If the need for consent was waived by the ethics committee, please include this information.

Reviewer's Responses to Questions

**Comments to the Author**

1. Is the manuscript technically sound, and do the data support the conclusions?

Reviewer #1: Yes

Reviewer #2: Yes

Reviewer #3: Yes

2. Has the statistical analysis been performed appropriately and rigorously? 

Reviewer #1: Yes

Reviewer #2: Yes

Reviewer #3: Yes

3. Have the authors made all data underlying the findings in their manuscript fully available?

Reviewer #1: Yes

Reviewer #2: Yes

Reviewer #3: Yes

4. Is the manuscript presented in an intelligible fashion and written in standard English?

Reviewer #1: Yes

Reviewer #2: Yes

Reviewer #3: Yes

5. Review Comments to the Author

Reviewer #1: The study is well conducted though it has limitations like factors responsible for C- section and others. Data analysis for the qualitative data would have made it stronger. Substantial efforts made on data collection and analysis.

Reviewer #2: The research submitted for review has been done meticulously and keeping the potential concerns in mind. Appropriate statistical analysis tools were used. The variables associated with the research have been largely covered. The language is easy to understand and has clarity.

Reviewer #3: This paper is addressing an important research question relating to the prevalence of C-section deliveries in North Cypress, time trends, and its association with breastfeeding rates. The authors used data from a representative cohort. The statistical analysis is robust with appropriate adjustments for important confounders. The manuscript is well-written, limitations are addressed, and implications for practice well-stated. I recommend acceptance.

6. PLOS authors have the option to publish the peer review history of their article (what does this mean?). If published, this will include your full peer review and any attached files.

Reviewer #1: No

Reviewer #2: **Yes: **Dr. Kandarp Mathur

Reviewer #3: **Yes: **Mona Nabulsi, MD, MS

---

## [Editor Report · Acceptance letter]

27 Jun 2023

PONE-D-22-23791 

Predictors and trends of Caesarean section and breastfeeding in the Eastern Mediterranean region: data from the cross-sectional Cyprus Women’s Health Research (COHERE) Initiative 

Dear Dr. Rahmioglu:

I'm pleased to inform you that your manuscript has been deemed suitable for publication in PLOS ONE. Congratulations! Your manuscript is now with our production department. 

Kind regards, 

on behalf of

Dr. Mona Nabulsi 

Academic Editor

PLOS ONE